# EFFECT OF TRAINING FRAGMENT LENGTH ON TRANSFORMERS IN TEXT COMPLEXITY PREDICTION

**Rafik Hachana, Vladimir Ivanov**
Innopolis University, Innopolis, 420500, Russia
{r.hachana, v.ivanov}@innopolis.ru

## ABSTRACT

With the myriad practical applications of text complexity classification, it is important to optimize the training text fragment size for performance. We experiment with fine-tuning pre-trained BERT models to predict the complexity of Russian school text using different fragment sizes for training.

## 1 INTRODUCTION

We aim to evaluate the effect of text fragment length on the performance and training time of Transformer models (Vaswani et al., 2017) in text complexity prediction. For this purpose, we fine-tune three pre-trained BERT models for the Russian language (Devlin et al., 2018) using a corpus of Russian school textbooks, by dividing the corpus with a different maximum fragment length each time. We measure the training time and MAE for each case. We also test the null hypothesis $H_0 = \{$Training fragment size is correlated with the model's MAE$\}$. The motivation is to find the best fragment length to optimize both training time and performance.

## 2 RELATED WORK

Text complexity prediction is a relevant task in Natural Language Processing. It has many practical applications, like curating texts for courses and textbooks (Fitzgerald et al. (2015), Kurdi (2020)). Some studies used morphosyntactic features to predict the text complexity (Kurdi (2020), Cuzzocrea et al. (2019)), while more recent studies such as Iavarone et al. (2021) and Alaparthi et al. (2022) use models based on BERT (Devlin et al., 2018). Martinc et al. (2021) ran a similar experiment to ours, comparing different models in the text complexity classification task, and found that BERT performed better on one dataset, and HAN (Yang et al., 2016) performed better on the 2 other used datasets. Iavarone et al. (2021) found that the usage of explicit language features performed better than BERT in a 7-class complexity classification task, and attributed this to the small size of the dataset, which is not enough to train a relatively large model like BERT.

## 3 METHODOLOGY

The experiment consists of creating a dataset of labeled text fragments from the corpus, then fine-tuning three pre-trained models on it. Each model is then evaluated using MAE. This experiment is repeated for different values of maximum allowed fragment length, ranging from 50 words to 510 words. To parse the text, we split it into words, then divide it into fragments of the desired fragment length in terms of words. The last fragment is kept as it is, even if it is shorter than the fragment length. Each fragment is labeled with a text complexity score. We use the corresponding school grade of the textbook to assign a text complexity score. The fragment labeling is further described in Appendix A.

For each fragment length, we fine-tune three BERT models using the corresponding fragment dataset to predict the complexity score. We picked three models pre-trained on Russian text: `rubert-base-cased` (RuBERT) , `rubert-base-cased-sentence` (RuBERT-s) , and `xlm-roberta-large-en-ru` (XLMR). The pre-training of the models is explained in Appendix B. The models are trained as regressors using the CLS token of the BERT output, which

Table 1: Correlation between fragment size and MAE

| Model | CE all | CE $> 100$ |
|---|---|---|
| RuBERT | $-0.35244494$ | $-0.11844252$ |
| RuBERT-s | $-0.43573392$ | $-0.30136838$ |
| XLMR | $-0.49109714$ | $-0.41563056$ |

is passed through a dense layer. All training parameters are described in Appendix C. We measure the training time for each model, and evaluate its performance using MAE. We also use the Pearson coefficient to assess the correlation between the fragment length and evaluation MAE. All Python code used for the experiment is available on our GitHub repository[1]

## 4 RESULTS

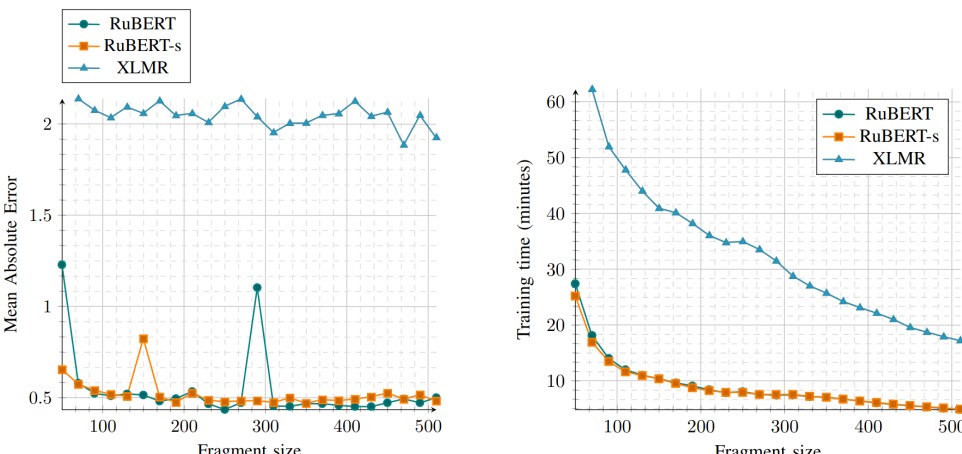

Figure 1: Left: MAE, Right: Model training time

In Figure 1, we illustrate the main results of the experiments. In terms of MAE, XLMR performed the worst, which we assume is due to model's need for more training epochs due to its size. However, the MAE does not significantly vary across different fragment sizes, except for sizes less than a 100 words where the performance seems to suffer more. We notice 2 unexplained performance drops at around 150 and 300 words, which we assume are spurious, but further experiments are needed. For the training time, we notice that it consistently decreases with larger fragments, which is explained by the fewer instances in datasets with larger fragments. The number of instances for each fragment size is illustrated in Appendix D.

We have also calculated the correlation between the MAE and fragment length for all models, either using all fragment sizes or only sizes larger than 100 words (Table 1). Using a correlation threshold of $\pm 0.6$, there is no correlation between fragment sizes larger than 100 words and MAE, therefore we reject the null hypothesis. However, there is a weak correlation when including smaller fragment sizes.

## 5 CONCLUSION

We experimented with fine-tuning BERT models on text complexity classification using different text fragment sizes. The results suggest that the MAE of the task is not affected by the fragment length, except for the smallest fragment sizes (less than 100 tokens), which implies that one can freely choose a fragment size for training without hindering the model performance. The training time is reduced with bigger fragment sizes without affecting performance.

---

[1]https://github.com/RafikHachana/school-text-complexity-classification

ACKNOWLEDGEMENTS

This research has been financially supported by the Analytical Center for the Government of the Russian Federation (Agreement No. 70-2021-00143 dd. 01.11.2021, IGK 000000D730321P5Q0002)

URM STATEMENT

The authors acknowledge that at least one key author of this work meets the URM criteria of ICLR 2023 Tiny Papers Track.

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

## A   TEXT COMPLEXITY ANNOTATIONS

We consider the complexity prediction task as a regression task and use the school grade as an indicator of the text complexity. Assuming that higher grades are correlated with more complex text passages, we assign a complexity score $C(x)$ to each text passage, with $x$ the school grade.

There are 11 grades in the Russian school system (numbered from 1 to 11). There are also "advanced" variants of the school grades (taught in pioneer schools). The advanced grades are only available for the 10th and 11th grade. To formulate the complexity score, we choose $C(x) = x$ for regular grades, and $C(x \text{ advanced}) = x + 0.5$ for advanced grades e.g. $C(10 \text{ advanced}) = 10.5$.

## B   PRE-TRAINED MODELS

In the experiment, we have used three pre-trained BERT models. Here is a more detailed explanation of their respective pre-training procedures:

- `rubert-base-cased` [2] : A transformer model using the BERT Devlin et al. (2018) architecture (12 layers, hidden size of 768, and 12 self-attention heads) initialized with the weights of the multilingual BERT, then trained on the MLM and NSP tasks using the Russian part of Wikipedia and news data. (Kuratov & Arkhipov, 2019)

- `rubert-base-cased-sentence` [3] : A sentence encoder for Russian. It is initialized with the weights of RuBERT (Kuratov & Arkhipov, 2019), and fine-tuned on the data from SNLI (Bowman et al., 2015) Google-translated to Russian, and on the Russian part of XNLI dev set (Conneau et al., 2018). The sentence representations here are mean-pooled from all the token embeddings, in the same manner as Sentence-BERT Reimers & Gurevych (2019).

- `xlm-roberta-large-en-ru` [4] : This model is an XLM-RoBERTa Large version reduced to the most frequent vocabulary in English and Russian. The original XLM-RoBERTa Large model (Conneau et al., 2020) model has 24 layers, a hidden size of 1024, 16 attention heads, for a total of 550M parameters. The model is trained on 2.5TB of filtered CommonCrawl data in 100 languages. The training is done on raw unlabeled text as in the original XLM-100 model (CONNEAU & Lample, 2019), and by applying optimization approaches suggested by Liu et al. (2019).

---

[2] https://huggingface.co/DeepPavlov/rubert-base-cased
[3] https://huggingface.co/DeepPavlov/rubert-base-cased-sentence
[4] https://huggingface.co/DeepPavlov/xlm-roberta-large-en-ru

## C    TRAINING PARAMETERS

We present all the used the training parameters for the fine-tuning of all models in Table 2.

Table 2: Training parameters for all models

| Parameter | Value |
|---|---|
| Loss function | MSE |
| Number of epochs | 3 |
| Training batch size | 5 |
| Evaluation batch size | 16 |
| Warmup steps | 500 |
| Weight decay | 0.01 |

## D    NUMBER OF TRAINING INSTANCES

Figure 2 shows the number of training text fragments for different text fragment sizes. This explains the faster training time for bigger fragments, since there are less training instances.

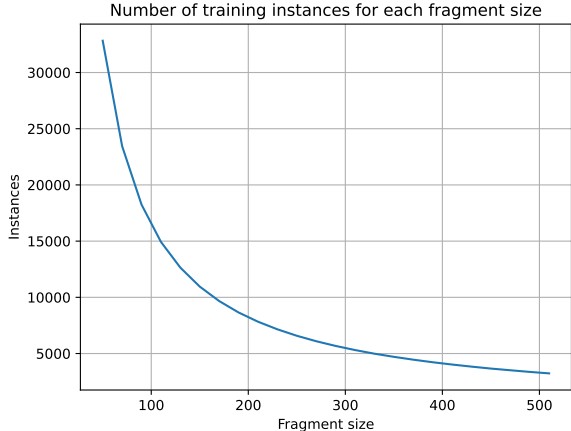

Figure 2: Number of training instances for each fragment size

