# OpenReview forum: "Effect of training fragment length on Transformers in text complexity prediction"
_ICLR.cc/2023/TinyPapers — Submitted to Tiny Papers @ ICLR 2023_

### Official Review · Reviewer_zbHS · 2023-03-23

**Confidence:** 5

**Summary Of Contributions:**

The paper proposes to use pre-trained models to predict text complexity in Russian and studies whether the length of the segment used during training/inference impacts complexity prediction.

**Rating:**

Great Start (GS): a submission which meets some of the reviewing criteria but has room for improvement

**Strengths And Weaknesses:**

Strengths: Text complexity prediction is an important task in NLP and is a preliminary step in identifying when and how a text should be simplified. This can help make texts more accessible to people with various reading or cognitive disabilities. The paper proposes to address this task in Russian.

Weaknesses: The paper lacks a clear description of the experimental setup: a) how the fragment length is computed (bpe-level/word-level) and b) how the pre-trained models (ex: xlm-roberta-large-en-ru)  were trained and c) how the representation is used to train the regressor as there are many different ways and that could also impact the results (for ex: CLS embeddings vs Mean-pooling for bert based models).

**Suggested Changes:**

Adding a clear motivation on the experimental design choices and the choice of hypothesis will help better understand and explain the results.

---

> ### Author Response · Authors · 2023-05-18
> **Response to Reviewer zbHS**
>
> Dear Reviewer,
>
> Thank you for going through our work. We have gone through the weaknesses and suggested changes mentioned in your comment. And we have also uploaded a new paper revision where we address all the issues with our first submitted manuscript.
>
> Regarding the ambiguities in the experimental setup:
> - how the fragment length is computed (bpe-level/word-level): We use whole words as a unit for fragment length. We have open-sourced all the code and data used and linked it in our latest revision.
> - how the pre-trained models (ex: xlm-roberta-large-en-ru) were trained: We have added an appendix with thorough details about the pre-training of each model used in the study.
> - how the representation is used to train the regressor as there are many different ways and that could also impact the results (for ex: CLS embeddings vs Mean-pooling for bert based models): We are linearly projecting the CLS embedding to get the final output of the regressor. We made sure to better describe the data annotation and the task description in the methodology and the appendices.
>
> And regarding the suggested changes:
> "Adding a clear motivation on the experimental design choices and the choice of hypothesis will help better understand and explain the results."
> -> We have explicitly stated that our motivation is characterizing the effect of different training fragment lengths on the performance and the training time. The ultimate goal is to optimize the training time without significantly hurting the performance of the model.
>
> We are eager to discuss any further concerns that you have about our paper. Thank you for your time and consideration.
>
> All the best,

---

### Author Response · Authors · 2023-05-17
**New paper revision**

Dear ICLR TinyPaper reviewers,

We express our gratitude for your thorough feedback on our study and for you constructive comments. We have taken the comments into consideration and we have uploaded a new paper revision with the following changes:
- We clarified some ambiguities that were pointed out by the reviewer, such as the method used to fragment the text and label the fragments (we use words to calculate the fragment size, not tokens with BPE), and how the regressor is trained (it is trained using the CLS token of the BERT output).
- We added multiple appendices to not miss any details about the experimental setup. The appendices discuss data annotation, the pre-trained models, training parameters and dataset size.
- We revisited the overall text and made it clearer and more concise.
- For reproducibility, we added a link to a public Github repository with the code and a link to the training dataset.
- We have explicitly stated the motivation behind the experimental setup in the introduction, the motivation is to measure the effect of fragment size on both the models evaluation performance as well as the training time in order to characterize the tradeoff between them, and eventually find an optimal text fragment size for both performance and training time.
- We have updated the title.

We think that we have covered all of the comments and requested changes of the reviewers. We hope that our new revision is on par with what is expected for an ICLR TinyPaper. We look forward for any further comments and suggestions from your side. Thank you for your time and consideration,

Best regards

---

### Author Response · Authors · 2023-05-30
**Paper archival**

Dear ICLR Program Chairs,

We have recently uploaded a new revision of our paper and replied to all reviewer comments.

We would like to state that we want to opt-in for the archival of our paper, in case our new revision is eligible.

All the best,
The authors

---

### Meta-Review · Area_Chair_EDPg · 2023-04-07

**Recommendation:** Invite to revise
**Confidence:** 5

**Metareview:**

The researchers fine-tuned three pre-trained BERT models (from Hugging face) on russian text, using Mean Absolute Error (MAE) as the primary evaluation metric. Their findings revealed that there was no significant correlation between text fragment size and the models performance.





**Summary:**

This study explored the impact of text fragment length on the performance of fine-tuned transformer-based models using Russian school textbook datasets.

**Reason For Not Giving A Higher Recommendation:**

The current paper lacks sufficient detail for reproducibility and does not clearly explain the formulation of the regression task, which is its main contribution. This lack of clarity and key information hinders the understanding and replication of the study.

**Reason For Not Giving A Lower Recommendation:**

NA

---

> ### Author Response · Authors · 2023-05-18
> **Response to Area Chair EDPg**
>
> Dear ICLR Area Chair,
>
> Thank you for considering our work. We have uploaded a new revision of our paper recently where we made multiple improvements and addressed all reviewer comments. We have listed all the improvements in our comments and made sure to cover all the concerns and questions of the reviewer.
>
> Regarding the limitations mentioned in your comment  ("The current paper lacks sufficient detail for reproducibility and does not clearly explain the formulation of the regression task, which is its main contribution. This lack of clarity and key information hinders the understanding and replication of the study.")
> -> We made sure to thoroughly explain the experimental setup of the regression task in the main text and in the appendix. We worked on improving the clarity of the paper overall, and we have also linked the code and data used for the experiment for reproducibility.
>
> We hope that the new version addresses all of your concerns. We look forward to addressing any further comments to ensure the best quality for our paper.
>
> All the best

---

### Decision · Program_Chairs · 2023-04-08

Revision accepted; invite to archive